# The Impact of Temporary Stay at High Altitude on the Circulatory System

**DOI:** 10.3390/jcm10081622

**Published:** 2021-04-12

**Authors:** Karolina Mikołajczak, Karolina Czerwińska, Witold Pilecki, Rafał Poręba, Paweł Gać, Małgorzata Poręba

**Affiliations:** 1Department of Pathophysiology, Wroclaw Medical University, Marcinkowskiego 1, PL 50-368 Wroclaw, Poland; karolina.mikolajczak@student.umed.wroc.pl (K.M.); witold.pilecki@umed.wroc.pl (W.P.); malgorzata.poreba@umed.wroc.pl (M.P.); 2Department of Hygiene, Wroclaw Medical University, Mikulicza-Radeckiego 7, PL 50-368 Wroclaw, Poland; karolina.czerwinska@umed.wroc.pl; 3Department of Internal and Occupational Diseases, Hypertension and Clinical Oncology, Wroclaw Medical University, Borowska 213, PL 50-556 Wroclaw, Poland; rafal.poreba@umed.wroc.pl

**Keywords:** environmental health, high altitude, circulatory system

## Abstract

In recent times many people stay temporarily at high altitudes. It is mainly associated with the growing popularity of regular air travel, as well as temporary trips to mountain regions. Variable environmental conditions, including pressure and temperature changes, have an impact on the human body. This paper analyses the physiological changes that may occur while staying at high altitude in healthy people and in people with cardiovascular diseases, such as arterial hypertension, pulmonary hypertension, heart failure, ischemic heart disease, or arrhythmias. Possible unfavourable changes were underlined. Currently recognized treatment recommendations or possible treatment modifications for patients planning to stay at high altitudes were also discussed.

## 1. Introduction

All over the world people inhabit diverse habitats located at different altitudes above sea level. It is estimated that about 140 million people permanently live above 2500 m above sea level (m a.s.l.), i.e., at high altitudes [1,2,3]. As many as 40 million people a year go on a temporary stay in such places [2]. Moreover, every year lots of people travel on planes, for example in 2019 it was as much as 4.543 billion passengers [4]. Passenger planes fly at an altitude of over 10,000 m a.s.l. [5]. Despite cabin pressurization, especially longer flights lead to gradual fall in cabin oxygen pressure resulting in the decrease in oxygen saturation in human bloodstream. Commercial aircraft are pressurized only to cabin altitudes of 1800–2500 m and it should be remembered that at 2438 m the partial pressure of oxygen falls giving even in healthy passengers the fall in arterial oxygen tension to between 8.0 and 10kPa (60–75 mm Hg) and oxygen saturation measured by pulse oximetry (Spo2) to 89–94%) [6]. Consequently, it is physiologically compensated by mild to moderate hyperventilation, simultaneously being limited by the fall in arterial carbon dioxide tension (Paco2), and a moderate tachycardia [6]. Relations between rising altitude and pressure as well as partial pressure of oxygen are presented in Figure 1, based on Samuels [7].

The definition of a mountain area varies from country to country. According to some European regulations mountain areas are considered to be places >2500 m or a combination of altitude and slope for lower areas, for example a minimum altitude of 1500 m and a slope above 2°, or an altitude of at least 1000 m and a steeper slope than 5°, or a minimum altitude of 300 m within a minimum of 7 km [8,9,10].

Based on the abovementioned definition, it can be assumed that mountain areas cover 33% of Eurasia, 19% of South America, 24% of North America, and 14% of Africa, which together constitute 24% of the Earth’s surface [11]. In the European Union, mountain areas cover more than 39% of the area; however, in some countries such as Switzerland and Norway this percentage is much higher and amounts to >70% [8].

Standard conditions at sea level are pressure 760 mmHg, temperature 15 °C and the fraction of oxygen in the air is 0.21 [12]. With high altitude, the barometric pressure decreases and results in lower levels of oxygen partial pressure. Barometric pressure drops gradually with increasing altitude. The temperature also decreases [12]. The same happens with absolute air humidity, which reaches low values. There is, however, high exposure to solar radiation [2,13]. The main factor influencing the functioning of the organism in such conditions is low atmospheric pressure and proportional reduction of oxygen partial pressure in the inhaled air [13,14,15]. Human body has to some extent an ability to adapt to changes in environmental conditions and the circulatory system is largely involved in the process.

This paper analyses the physiological changes that may occur while staying at high altitude in healthy people and in people with cardiovascular diseases, such as arterial hypertension, pulmonary hypertension, heart failure, coronary artery disease, and arrhythmias. Possible treatment modifications before the planned high-altitude stay were also considered. The presented data concern acute (3-day) and sub-acute (up to 14 days) exposure [3]. The work does not address the issue of chronic exposure.

The profound surveys analyzing relations between staying at high altitude and cardiovascular problems employing not only the healthy travelers, but also people with pre-existing heart and vessel diseases have been scarce so far and such reviews are rarely available and the aim of this article is to fill the gap in this subject.

## 2. Methods

The literature analysis was performed using PubMed and Google Scholar databases including years July 1977 to February 2021. All types of articles were included such as reviews, meta-analyses, case reports, and randomized controlled trials; however, it should be underlined that out of the last group of randomized clinical trials only few have been found in this field. The following keywords were used: high altitude; acute mountain sickness, circulatory system, parameters at high altitude, arterial hypertension at high altitude, pulmonary hypertension at high altitude, heart failure at high altitude, coronary artery disease at high altitude, arrhythmia at high altitude, and ischemic stroke at high altitude.

## 3. Physiology and Pathophysiology in Healthy at High Altitude

### 3.1. Normal Physiological Adaptation

The human body must adapt to the conditions at high altitude. This process is called acclimatization. It consists of a series of physiological processes, occurring in lowlanders, which are beneficial in a hypoxic environment [12].

Increasing ventilation is the first acclimatization mechanism [13]. It is caused by the stimulation of peripheral receptors and an increase in the activity of the sympathetic system [16]. Hyperventilation leads to a decrease in the CO2 concentration in the alveoli and hence in the blood. Hypocapnia together with alkalosis affect the respiratory center and inhibit it, which prevents excessive ventilation [13]. The kidneys also contribute to the maintenance of the acid–base balance by removing excess bicarbonate and preserving hydrogen ions [13].

As a result of staying at high altitude, pressure in the pulmonary arteries increases significantly [17]. This is due to hypoxic pulmonary vasoconstriction. As a result, the afterload in the right ventricle increases, which in turn reduces the volume of blood returning to the left ventricle [18].

When at high altitude, initial changes in the systemic circulation include tachycardia and an increase in cardiac output; however, the stroke volume does not change; the main physiological and pathophysiological changes in high altitude conditions are presented in Table 1 [17].

After 3–5 days, the heart rate is still increased, but the cardiac output returns to normal, which is a consequence of the decrease in stroke volume [14,17,18,28]. The mechanism of stroke volume reduction is not fully understood [18,28]. One of the probable factors is the reduction of end-diastolic volume caused by reduced plasma volume; although, this has not been confirmed [14,18]. In acclimatized people, the cardiac output is the same for a given workload, both at high altitude and at sea level [18,28]. However, the maximum cardiac output is reduced [17,18].

Opinions differ on the issue of ventricular contraction. According to Naeije R., ventricular systolic function is at first increased and then maintained or slightly reduced [17]. However, according to Bilo G. et al. there are no major changes in left ventricular contractility; although, there are changes in mechanics that may be due to decreased oxygen supply to the subendocardial layers of the heart muscle [14].

Rapid climbing to high altitude in healthy subjects is, as recently reported, associated with increased right atrial contractility which is a consequence of increased right ventricular afterload [29]. It has been reported that exposure to high altitude impairs the function of both atria, but especially the contractile function of the right atrium [30].

During the first days of being at high-altitude blood pressure increases [3,17]. The increase may be noticeable 24 h a day, but at very high altitudes (>5400 m a.s.l.) it occurs mainly at night [14]. It is the result of the interaction of various mechanisms, including increased activity of the sympathetic system, release of endothelin, increased aortic stiffness, impaired endothelial function, and increased blood viscosity [3,14]. Moreover, the increase in erythropoietin concentration after 16 h at high altitude may be one of the factors contributing to the increase in blood pressure [3]. Another mechanism that contributes to the adaptation is the renin–angiotensin–aldosterone system (RAAS). Angiotensin acts on skeletal muscles and the heart, and causes sodium and fluid retention, which results in an increase in blood pressure [31]. It has been shown that angiotensin levels increase with increasing altitude and decrease with decreasing altitude [32]. Interestingly, during exercise performed under hypoxic conditions, the close relationship between renin and aldosterone seems to diminish because of a decrease in the activity of angiotensin-converting enzyme (ACE—Angiotensin-converting Enzyme) [33]. In the initial phase of physical activity, there is an increase in plasma renin activity (PRA) and plasma aldosterone concentration (PAC) and ACE activity remains constant [33]. However, with the continuation of exercise and increasing hypoxia, there seems to be a further increase in PRA but a decrease in PAC and ACE [33]. This may be due to the activation of angiotensin II-degrading enzymes and/or a decrease in the density of angiotensin II receptors in the adrenal cortex, which protects against an excessive increase in aldosterone concentration and, consequently, an excessive increase in blood pressure in response to PRA [34].

### 3.2. Acute Altitude Illnesses

The main problem that may arise when climbing to high altitudes without sufficient acclimatization is acute high altitude illnesses. It applies to both sick and those considered healthy [35]. The first symptoms usually appear after 6–12 h of staying above 2500 m a.s.l. [36]. In such a situation, people exposed to high altitude and extreme conditions associated with it may develop dyspnea during exercise as well as during rest, also cough, nausea, fatigue, headaches, sleep problems and changes in mental state [37]. Decreased sleep quality was reported during the first few days spent at high altitude; however, this problem requires further study [38]. Moreover, in new acute mountain sickness scoring sleep question have been eliminated as it improves the diagnosis of true AMS [39].

Subsequently, the disease may take one of three forms: acute mountain sickness, high altitude cerebral edema (HACE) or high altitude pulmonary edema (HAPE) [37,40].

The initial cause is hypoxia. Hypoxia leads to hypoxemia, which causes the following reactions: increased capillary pressure (as a result of increased cerebral blood flow), increased cerebral blood volume and increased permeability of blood–brain barrier [41]. This has the effect of brain swelling and consequently also inadequate buffering by cerebrospinal fluid [41]. This can cause AMS, which can progress to HACE [41]

Recently, it has been suggested that there may be genomic associations with susceptible in some individuals to high altitude illness proposing that single nucleotide polymorphisms (SNPs) are involved in the genesis of AMS; moreover, EPAS1 and VEGFA gene variants were found to have such a relation [42]. Additionally, authors suggest that this tool might be useful for screening susceptible populations and predicting clinical symptoms in future [42].

Alveolar hypoxia along with sympathetic overactivity, endothelial dysfunction, cold, and exercise leads to raised pulmonary hypertension, which in turn causes increased capillary pressure that trigger endothelial stress [41]. Then, endothelial stress together with inflammation and decreased alveolar clearance of sodium and water causes capillary leakage [41]. The probable cause is capillary leakage in the brain leads to AMS and HACE or if it is in the lungs to—HAPE [40]. Nevertheless, the pathophysiology of these phenomena is not yet well understood.

AMS can present as one of three forms: mild, moderate, or severe. Symptoms characterizing this condition are included into four groups: headache, gastrointestinal type (nausea, vomiting, depending on its severity), fatigue and/or weakness, and dizziness/light headedness [39]. In mild form a patient has 3–5 points from the scoring system, and in the severe form, the result of 10–12 points [39].

HACE can develop from AMS. It may be accompanied by symptoms such as truncal ataxia, decreased consciousness, mild fever, and drowsiness. [43]. HAPE, on the other hand, is characterized by inappropriate dyspnea during exercise, dry cough with exertion, reduced exercise performance, pink sputum, and drowsiness [43].

In a new relatively large-scale case-control study it was revealed that the decrease in stroke volume index correlated with the altered left ventricular filling pattern was related to the onset and severity of AMS [44].

### 3.3. Clinical Recommendations

The most effective treatment is to go down immediately and take oxygen [45]. In the case of acute mountain sickness, acetazolamide, or dexamethasone can be used, in cerebral edema—dexamethasone and in pulmonary edema—nifedipine [43,45]. The simplest method of prophylaxis is slow climbing to the higher areas [37]. The recommended climbing speed is 300 to 500 m per day separated by a rest day every 3–4 days [43]. Administration of acetazolamide and dexamethasone can also be effective in prophylaxis of AMS and HACE [43,45]. Acetazolamide can be used in doses 125, 250, and 375 mg/bid [46]. Ibuprofen may be applied in the prevention of AMS in case of allergy or intolerance to acetzolamide or dexamethasone or in persons who do not wish to take those drugs [45]. Administration of budesonide is also theoretically an effective prophylaxis for the prevention of mild AMS, but it is not effective with the prevention of severe AMS [47]. However, not all authors support the idea of the beneficial role of budesonide in this case [48]. Both papers agree that its use does not cause side effects [47,48]. Caution should be taken however, as further research is needed in this area [47]. To avoid HAPE nifedipine, tadalafil, or sildenafil can be potentially used [45].

## 4. Patients with Cardiovascular Diseases at Higher Altitudes

### 4.1. Arterial Hypertension

#### 4.1.1. Physiology and Pathophysiology

The prevalence of arterial hypertension in the world varies, however ranges to about 1.13 billion of the adult population [49]. In Poland, the percentage of patients among adults was, depending on the study, 29% (NATPOL PLUS) or 45% (WOBASZ II), and among the elderly (aged 65 and over)—75% (PolSenior) [50]. Many of these patients choose to travel by air or visit a high-altitude location, which exposes them to altered environmental conditions.

Among people not burdened with arterial hypertension, both systolic and diastolic blood pressure increase after several hours of staying at high altitude (Table 1) [17]. Bilo G. et al. indicate that in patients with arterial hypertension this increase is even higher [1,51]. Therefore, it seems that these patients are at a higher risk of potential cardiovascular complications and require adequate blood pressure control. Interestingly, Duke et al. noted that the presence of hypertension does not affect the risk of AMS and may even be associated with a reduction of this risk, although more research is needed on this topic [52].

#### 4.1.2. Clinical Recommendations

Some medications that are used at low altitudes may not be fully effective in patients staying at high altitudes. This applies to such groups of pharmaceuticals as: non-selective beta-blockers and angiotensin receptor blockers (ARB) [53,54]. The disadvantage of using non-selective beta-blockers is that they reduce the oxygen saturation of hemoglobin, thus limiting the ability to exercise. The use of selective beta-blockers does not generate such effects; the clinical recommendations are presented in (Table 2) [53]. The downside of using long-acting ARBs, e.g., telmisartan, at an altitude of 5400 m a.s.l. is the limitation of the renin–angiotensin–aldosterone system. At altitudes below 3400 m a.s.l. telmisartan seems to maintain its effectiveness [54]. It was also shown that in high mountain conditions it is beneficial to treat mild hypertension with a combination of a calcium channel blocker (CCB) and ARBs [51]. Caravita et al. conducted a double-blind randomized trial involving patients with mild hypertension. Patients received placebo or 80 mg of telmisartan and 30 mg of slow-release nifedipine. Use of these drugs reduced hypoxia-driven upward shift and steepening of the blood pressure response to exercise and improved muscle oxygen supply [55]. Moreover, in people professionally involved in flying (e.g., pilots), Altaktazid (spironolactone combined with hydrochlorothiazide) also turned out to be effective in the treatment of arterial hypertension [56]. Side effects included a moderate loss of body weight and plasma volume, as well as a slight deterioration in renal function. This pharmaceutical is recognized as a safe and effective second-line drug for aircraft crew members [56]. The most important thing is that patients should monitor their blood pressure regularly and, if necessary, change the treatment to the one previously agreed with the doctor [1,57].

### 4.2. Pulmonary Hypertension

#### 4.2.1. Physiology and Pathophysiology

For people with pulmonary hypertension being at high altitudes could be dangerous. In such conditions of low pressure in response to hypoxia pulmonary blood vessels constrict [6]. The result is an overload of the right heart [19]. The aircraft cabin pressure is regulated, so healthy travelers should not feel difference, but passengers with pulmonary hypertension may experience an increase in pulmonary artery pressure [6].

#### 4.2.2. Clinical Recommendations

The safety of air travel among patients with pulmonary hypertension depends on the assigned NYHA class—the class of heart failure according to the New York Heart Association [19]. People with functional classes I and II can travel without oxygen supply, while patients with NYHA III and IV should have it provided [19]. The need to ensure access to oxygen also applies to all patients with PaO2 below 60 mmHg [1]. Severe pulmonary hypertension is one of the contraindications for air travel (pneumothorax and bronchial cysts are others) [58]. In those patients, rising to high altitudes is associated with a greater risk than in people suffering from diseases such as hypertension, coronary artery disease or bronchial asthma [36]. People with pulmonary hypertension should have access to oxygen both during flight and during trips to altitudes above 1500–2000 m a.s.l. [1,19]. At the moment, however, it is impossible to state which of the patients will have the need to use it during the trip [19].

### 4.3. Heart Failure

#### 4.3.1. Physiology and Pathophysiology

In Poland, about 1.39 million people suffer from heart failure [59]. In the USA, it is as many as 6.2 million adults [60]. Worldwide, the problem affects 64.3 million people [61]. Many scientific studies indicate that staying at high altitude may be considered safe for these patients, however, with a few reservations [1,62,63].

It should be remembered that in people with heart failure, the decline in physical fitness with increasing altitude is greater than in healthy people [63]. This decrease reaches 4% (in case of slightly reduced exercise capacity) or 10% (in case of significantly reduced exercise capacity) for every 1000 m [63]. In healthy people it is 8% for every additional 1000 m of altitude above 700 m a.s.l. up to 6300 m a.s.l. [64]. Patients should also avoid hypoxia, because they may experience worsening of symptoms due to increased neurohormonal activity [6]. A significant issue is the frequent coexistence of other diseases, such as pulmonary hypertension, chronic obstructive pulmonary disease (COPD) or ischemic disease, which reduce adaptive abilities [63]. According to Furian et al. patients with moderate to severe COPD experience a 54% reduction in exercise capacity at 2590 m, compared to 490 m, which can impede daily life activities at high altitude [65].

#### 4.3.2. Clinical Recommendations

The issue of air travel for people with cardiovascular diseases is regulated by the Canadian Cardiovascular Society (CCS) recommendations [25]. Patients classified as NYHA I and II can use air transport without restrictions. Oxygen may be required in patients classified as NYHA III [25]. The British Thoracic Society (BTS) guidelines present a slightly different view. According to them patients with NYHA I, II, and III can travel without oxygen, but it is necessary for patients with NYHA IV, who can travel only in exceptional situations. Then a dose of 2 mL/min should be used [6]. Patients with hypoxemia at sea level or with coexisting lung diseases should be qualified for hypoxic challenge test (HCT) before flight [6].

However, it is worth remembering that these are not the only guidelines and there are other, often contradictory [66]. A different position from CCS is presented by Aerospace Medical Association (AsMA), UK Civil Aviation Authority’s Aviation Health Unit (AHU), American College of Cardiology/American Heart Association (ACC/AHA) and British Cardiovascular Society (BCS) [66]. This paper, however, mainly presents the position of CCS and BTS.

Inadequately selected treatment may be harmful to patients. The drug classes of beta-blockers, ACEI (angiotensin-converting-enzyme inhibitors), and ARB are effective in controlling heart failure at sea level. Unfortunately, the side effects of their use are much more unfavourable at high altitudes [1,63]. That is because they limit the physiological adaptation of the organism to high mountain conditions. ACEI and ARB reduce renal secretion of erythropoietin and limit hematocrit growth. ACEI and nonselective beta-blockers by acting on adrenergic receptors (mainly β2) reduce both gas diffusion in the alveoli and hyperventilation caused by hypoxia [1,63]. For this reason, it is advisable to replace non-selective beta-blockers (such as carvedilol) with selective β1-blockers (such as nebivolol, and bisoprolol) during a temporary stay at high altitude [63]. Moreover, in patients staying at high altitudes, carbonic anhydrase inhibitors, e.g., acetazolamide, may be used, but their combination with other groups of diuretics is not recommended, as it increases the risk of dehydration and electrolyte disturbances [1].

### 4.4. Coronary Artery Disease (CAD)

#### 4.4.1. Physiology and Pathophysiology

Adaptation to high altitude in patients with coronary artery disease may be difficult. In a hypoxic environment, cardiac output increases to maintain a constant supply of oxygen to the tissues. In CAD patients atherosclerotic plaques in the arteries prevent their dilatation, which causes reduced oxygen supply to the heart and may lead to acute coronary syndrome [1]. However, on analyzing the impact of the altitude on patient with CAD numerous mechanisms should be not forgotten such as: the possibility of exacerbating symptoms by reduced oxygen availability, the impact of hypoxia and other associated environmental conditions including exercise, dehydration, thermal stress, emotional stress, which may theoretically precipitate acute coronary events, and, eventually, in cases where patient is older and unfit, and did not have the sufficient acclimatization, angina also may occur [67].

#### 4.4.2. Clinical Recommendations

Coronary artery disease is not an absolute contraindication to staying at high altitudes [36]; however, each case should be considered individually. The safety of staying at high altitudes among this group of patients depends on several factors. People who develop angina during light effort at sea level should not climb to great altitudes. This is due to the fact that symptoms may be exacerbated in the hypoxic environment [68]. Likewise, patients after a recent heart attack should not stay at high altitudes. On the other hand, it seems that patients with stable angina pectoris and with a history of asymptomatic myocardial infarction (MI) may stay at high altitude without increased risk [68]. According to Schmid JP. et al. if the myocardial infarction occurred more than six months ago, and the patient was revascularized and classified as low-risk, despite the significant increase in oxygen demand and the elevation of lactate levels, the patient is capable of submaximal exercise at high altitude [69]. Patients with angina and classified as NYHA I and II have no contraindications for using commercial airlines. Patients with NYHA III may require oxygen supplementation, and patients with NYHA IV—should fly only when it is medically necessary and under the supervision of a physician [25]. A drug that can be used in patients with ischemic heart disease at high altitudes is acetazolamide, which compensates for the reduction in oxygen supply to the heart muscle [1].

Thomas MD. et al. attempted to assess the safety of air transport in patients after a myocardial infarction [70]. The study group consisted of 213 people transported by commercial airlines after a myocardial infarction in a foreign country. Safe repatriation was the purpose of the transport. Statistical analysis of the collected data showed that a heart attack even less than 14 days earlier is not a contraindication to the flight, provided that the patient is classified as NYHA I [25,70]. There were cases of asymptomatic hypoxia among the transported patients, but it was correctable with oxygen administration. There were also cases of angina pectoris among patients, but symptoms resolved after sublingual administration of nitroglycerin. In the abovementioned study, no significant differences were observed between the safety of transporting patients after ST-elevation myocardial infarction (STEMI) and non-ST segment elevation myocardial infarction (NSTEMI). The fact of undergoing revascularization before the flight also did not seem to affect the safety of transport. Air transport seems to be safe for patients, provided that their clinical condition is stable and that theoretically, medical care is provided during the flight [70].

### 4.5. Arrhythmias

#### 4.5.1. Physiology and Pathophysiology

The factors associated with high altitude that predispose to myocardial ischemia and hence to arrhythmia include increased activity of the sympathetic nervous system, hypoxia, right ventricular overload, alkalosis, and changes in the transmembrane potassium transport [1,24]. From the above, the most frequently discussed factor is sympathetic stimulation [20,21,22,23]. It increases the risk of sudden cardiac death, especially in patients after MI [20]. As much as 30% of deaths while practicing alpine sports are caused by sudden cardiac death [24].

Kujaník S. et al. indicated that the incidence of supraventricular premature beats (SVPB) and ventricular beats (VEB) in healthy persons at the moderate altitude (1350 m a.s.l.) was twofold higher and its periodicity is shifted when compared with the low altitude of 200 m a.s.l. [22]. In this view, it should be presumed that in patients with pre-existing cardiovascular diseases, the incidence of different types of arrhythmias at higher altitude may be more evident, even though we lack large-scale studies in this area.

There are, however, scientific reports presenting contrary findings. Woods DR. et al. indicate that the risk of dangerous arrhythmias in people prepared to exercise at high altitudes is low [24]. However, tachycardia that increases with altitude is common [24]. With increasing altitude, patients may experience palpitations, not only when exercising but, less frequently, at rest [24].

Regarding QTc interval prolongation, the findings are also inconclusive. According to the study by Carta et al. the QTc time does not change significantly during ascent [71]. In contrast, Bisang’s results indicate a significant prolongation of QTc during nocturnal measurements [72]. Both mentioned studies were conducted on different groups of patients; however, all had moderate to severe COPD [71,72]. Furthermore, in another study in 24-Holter monitoring of an unacclimatized 65-year-old man without pre-existing heart disease during a climb to 5895 m QT remained unchained, only the ventricular arrhythmia in the form of attacks of ventricular bigeminy occurred [73].

#### 4.5.2. Clinical Recommendations

For patients already diagnosed with arrhythmia, the issue of traveling to the high-altitude locations needs more considerations. It appears that patients with paroxysmal atrial fibrillation may travel safely to high altitudes [16]. They are no restrictions for patients with well-controlled supraventricular arrhythmias classified as NYHA I or NYHA II [25]. However, it is contraindicated in people with unstable arrhythmia and ventricular arrhythmia classified as 4b in the old Lown scale, which means the presence of non-sustained ventricular tachycardia of various duration [16]. People with uncontrolled hemodynamically significant ventricular arrhythmias classified as NYHA III or NYHA IV should not travel by commercial airlines [25].

The occurrence of arrhythmias is one of the main disqualifying factors for the pilot profession in Western Europe [74]. People who want to work as aircraft personnel must have, among others, 12-lead electrocardiography (ECG) and serum lipid measurement. Other tests that may be indicated are: 24-h Holter ECG, myocardial perfusion imaging, MRI, coronary angiography, and many others. If the applicant is found to have any abnormality of the cardiovascular system that might affect flight safety, he or she shall require cardiological consultation and an appropriate evaluation. The most common disabling arrhythmias are premature ventricular beats, ventricular tachycardia, and paroxysmal atrial fibrillation [74].

### 4.6. Peripheral Circulatory Disorders

#### 4.6.1. Physiology and Pathophysiology

Lower-limb oedema is the most common ailment faced by patients flying long distances (over 7 h) [75]. This problem may affect even 86% of travelers [76]. The causes of edema formation are increased capillary filtration, reduced cabin pressure, reduced space and mobility, and a change in fluid balance during flights. The above changes are physiological and differ in their severity from one patient to another [25].

Air travel as well as mountain climbing are both risk factors for venous thromboembolism (VTE). When in the high mountains people are exposed to some factors that affect blood hypercoagulability; and these include hypoxia, dehydration, low temperature, and the use of tight-fitting clothing [77]. As for the frequency of symptoms of deep vein thrombosis, it ranges from 0.0% to 0.28%, and is asymptomatic between 0.0% and 10.34% [78]. In another study, a similar result was obtained, and it was 0.3% [79]. Long journeys greater than 8 h are associated with the risk of developing VTE in the form of deep vein thrombosis (DVT) or pulmonary embolism (PE) [80]; however, air travel of less than 8 h does not appear to affect the risk of DVT [78,81]. VTE risk factors include recent surgery, a history of VTE, pregnancy, puerperium, hormone replacement therapy, active neoplastic disease, obesity the use of oral contraceptives, neoplastic diseases, and thrombotic states [80,81,82]. Opinions and recommendations as to whether a stay at high altitude affects the development of VTE are often contradictory [77]. According to Cancienne JM. et al. there is a relationship between high altitude stay and VTE. In the conducted case-control study, they found that among patients who underwent elective shoulder arthroscopy at an altitude of more than 1219 m a.s.l. (4000 feet), the incidence of VTE was higher than in patients who underwent the same procedure below about 30 m a.s.l. (100 feet) (OR, 2.6; *p* < 0.0001) [83]. However, further research should be conducted to clarify this point.

#### 4.6.2. Clinical Recommendations

Oral administration of o (beta-hydroxyethyl)-rutosides was shown to be an effective method of combating these ailments [75,76]. They prevent changes in skin perfusion (reduce the increased capillary filtration), thus inhibiting the development of edema [74,75]. The greater the dose, the smaller the swelling [75]. According to some reports, they can also be used in healthy people prone to edema, for economic reasons and a low risk of side effects. Compression therapy is also a frequently recommended procedure [75].

According to the American Society of Hematology Guidelines, people without VTE risk factors traveling long distances (over 4 h) should not use antithrombotic prophylaxis [80]. In patients with an increased risk of thrombosis, the use of graded compression stockings or standard dose low molecular weight heparin is suggested, and if it cannot be used—the use of acetylsalicylic acid (ASA) is recommended (Table 2) [80].

A biomarker that may be used in future for testing of patients who develop high altitude deep vein thrombosis is lncRNA (long non-coding RNAs), as suggested in novel studies by Jha et al. [84].

### 4.7. Ischemic Stroke

#### 4.7.1. Physiology and Pathophysiology

There is no consensus on the relationship between the frequency of stroke and being at high altitude. Some scientific studies indicate that stroke occurs more often in people staying at high altitudes [26,27]. This may be due to hypoxia and dehydration [26]. Others, however, indicate that among patients after stroke, regardless of the altitude, the adhesiveness of platelets is equally increased in patients staying at altitudes below 1000 m a.s.l. and above 3000 m a.s.l. [85]. Polycythemia is an important risk factor increasing the likelihood of stroke [27]. Interestingly, even among patients diagnosed with symptomatic stenosis of the carotid artery and the resulting cerebral blood supply disorders, there is no contraindication to flying [86].

#### 4.7.2. Clinical Recommendations

However, caution should be taken when diagnosing stroke in patients without risk factors, who have recently returned from an alpine expedition and have neurological symptoms characteristic of the disease. Similar clinical symptoms may occur in the course of high altitude cerebral edema (HACE), which must be considered in the differential diagnosis. This is important as stroke is treated with thrombolysis, and HACE with dexamethasone. If appropriate treatment is administered quickly enough, there is a chance of a complete cure of the patient [87].

Limitation of the study is a result of the understandable fact that, despite applying numerous publications, the randomized trials in this area, as aforementioned, are rather scarce. The available data were discussed; however, more research is necessary to make stronger support for some medical recommendations, especially knowing that currently lots of people travel regularly and visit places at high altitudes, also frequently use plane flights as a mode of transport. In this aspect, the clinical recommendations, although created by the prominent medical associations, should be treated individually for every patient and the doctors should better stay all the time open-minded to attain new knowledge in future.

## 5. Conclusions

When staying at high altitudes numerous physiological changes take place in the human body. These changes are essential to adaptation and functioning in the environment of hypobaric hypoxia. Some of them, however, can pose a risk to patients with chronic diseases, especially of the cardiovascular system. Still today, we lack the knowledge whether low altitude treatments are effective in preventing altitude-related worsening of disease-specific symptoms. Before the planned travel to a high-altitude location or a plane travel, such patients should consult a doctor to determine if there is a need for treatment modification. Similarly, it is for patients who plan a trip in the mountains and additionally, they should be educated about the rules of treatment in case of the acute mountain sickness.

## Figures and Tables

**Figure 1 jcm-10-01622-f001:**
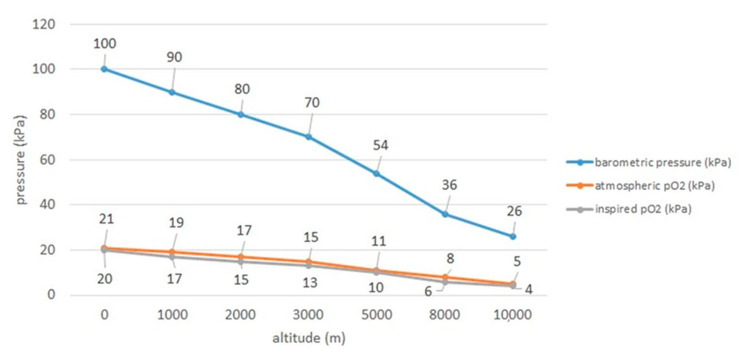
Relations of pressure and oxygen with rising altitude based on Samuels 2004 [7].

**Table 1 jcm-10-01622-t001:** Main physiological and pathophysiological changes in high altitude conditions.

Physiological Changes	Pathophysiological Changes
✔ Increased ventilation [13] ✔ Hypocapnia [13] ✔ Increased pressure in the pulmonary arteries [17] ✔ Increased activity of the sympathetic system [16] ✔ Tachycardia [17] ✔ Increased blood pressure [3] ✔ At first stage—the increase then the decrease of stroke volume [17] ✔ Reduced plasma volume [14,18] ✔ Increased blood viscosity [3,14] ✔ Increased erythropoietin concentration [3]	✔ Reduced oxygen supply to the heart—in patients with CAD as atherosclerotic plaques in the arteries prevent their dilatation [1] ✔ Contraction of the pulmonary vessels and overload of the right heart in patient with pulmonary hypertension [6,19] ✔ Increased activity of the sympathetic nervous system [20,21,22,23] ✔ Increased neurohormonal activity—dangerous in patients with heart failure [6] ✔ Higher increased systolic and diastolic blood pressure in patients with arterial hypertension [17] ✔ Predisposition to arrhythmia due to hypoxia, right ventricular overload, alkalosis, and changes in the transmembrane potassium transport [1,24] ✔ Edemas due to increased capillary filtration, reduced cabin pressure, reduced space, and mobility [25] ✔ Predisposition to ischemic stroke due to polycythemia, hypoxia, and dehydration [26,27]

CAD—coronary artery disease.

**Table 2 jcm-10-01622-t002:** Suggested therapeutic recommendations for patients with cardiovascular diseases at high altitudes.

Disease	Therapeutic Options
Arterial hypertension	Replacement of non-selective beta-blockers for selective beta-blockers [52]—randomized controlled studies I C recommendation Angiotensin II receptor blockade (suggested Telmisartan) lowers BP in healthy subjects up to 3400 m—randomized controlled studies IB recommendation Regular blood pressure measurements [1]—randomized controlled studies IIA recommendation
Pulmonary hypertension	Access to oxygen during flight and trips to altitudes above 1500–2000 m a.s.l. [1,57] III and IV class patients should avoid exposure to altitudes > 2000 m, and the access to oxygen supplementation if exposed to altitudes 1500-2000 m randomized controlled studies IC recommendation
Heart failure	Replacement of non-selective beta-blockers for selective β1-blockers [1]—randomized controlled studies IB recommendation Acetazolamide [1]—among diuretics may be considered mentioned in recommendations, experts’ opinion NYHA I, NYHA II, NYHA III—patients can fly without oxygen [6]—Well-conducted case-control or cohort studies with a low risk of confounding or bias and a moderate probability that the relationship is causal NYHA IV—patients should fly only in case of medical necessity [6]—Well-conducted case-control or cohort studies with a low risk of confounding or bias and a moderate probability that the relationship is causal
Coronary artery disease	Continuation of the previous therapy at high altitude (1) randomized controlled studies IC recommendation NYHA I, NYHA II—patients can fly with commercial airlines [65]—Experts’ opinion NYHA III—patients may require oxygen supplementation [65]—Experts’ opinion NYHA IV—patients can fly only, when it is necessary [65]—CCS Consensus Conference
Arrhythmia	They are no restrictions for patients with well-controlled supraventricular arrhythmias classified as NYHA I or NYHA II, [65]—randomized controlled study IIA recommendation uncontrolled hemodynamically significant ventricular arrhythmias classified as NYHA III or NYHA IV—should not travel by commercial airline [1,65]—randomized controlled study IIC recommendation
Peripheral circulatory disorders	In patients with oedema caused by venous hypertension flying for more than 7 h—oral administration of beta-hydroxyethyl-rutosides [73,75]—prospective, randomized, controlled trials In patients without VTE risk factors—no need for thromboprophylaxis [79]—based on systematic reviews and meta-analyses In patients with increased risk of thrombosis—graded compression stockings or standard dose low molecular weight heparin [79]—based on systematic reviews and meta-analyses If heparin cannot be used—acetylsalicylic acid (ASA) [79], based on systematic reviews and meta-analyses
Brain vessel diseases	Trekking or hiking at high altitude ≤3 months after stroke or TIA should be avoided -randomized controlled study IC recommendation [1] Stenosis of the carotid artery and the resulting cerebral blood supply disorders—is no contraindication to flying [85]—case-control study HACE should be treated with dexamethasone; thrombolysis is not recommended (only in case of ischemic stroke) [86]—case report

VTE—venous thromboembolism, BTS—British Thoracic Society, CCS—Canadian Cardiovascular Society, NYHA—New York Heart Association, HACE—high altitude cerebral edema.

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
