# Peer review of "The Impact of Temporary Stay at High Altitude on the Circulatory System"

_jcm, 2021, doi:10.3390/jcm10081622_

Round 1

Reviewer 1 Report

The introduction and the paragraph on the physical characteristics of the mountain are appropriate, although I would have better described the reduction of pressure and oxygen even more with the use of table or graph.

I found paragraph 4 on the lung a bit confusing. Rows from 108 to 111 are general and not related to the lung. I would have created a specific paragraph for AMS.

Paragraph 5.2: the claim concerning negative pressure in the aircraft cabin isn't correct.

Paragraph 5.5: the part concerning the arrhythmias time onset is not relevant for clinical advice purposes.

Paragraph 5.6: there are repetitions regarding DVT risk factors.

Paragraph 5.7:  High Altitude Cerebral Edema=HACE and NOT HACO!!!

In general I would better separate physiological and physiopathological parts from clinical and therapeutical advices, also in this case using tables.

Author Response

Reviewer 1:

Answering reviewers and their precious and constructive comments we have attempted to re-arrange body of the text taking all points into account. The manuscript has been thoroughly corrected of all errors and new suggested parts have been included, also we have attached the suitable references. Also, some unclear sentences were erased or replaced by the new ones addressing the problem properly. Major changes are listed below:

The introduction and the paragraph on the physical characteristics of the mountain are appropriate, although I would have better described the reduction of pressure and oxygen even more with the use of table or graph.

We created a special table showing relations of pressure and oxygen at altitude based on literature.

I found paragraph 4 on the lung a bit confusing. Rows from 108 to 111 are general and not related to the lung. I would have created a specific paragraph for AMS.

We have created a special paragraph on acute mountain sickness (AMS).

Paragraph 5.2: the claim concerning negative pressure in the aircraft cabin isn't correct.

We corrected in Paragraph 5.2 the description of pressure in the aircraft cabin specially that both Reviewers underlined the problem so it has been done in the ways suggested.

Paragraph 5.5: the part concerning the arrhythmias time onset is not relevant for clinical advice purposes.

In Paragraph 5.5 the part concerning the arrhythmias time onset was deleted.

Paragraph 5.6: there are repetitions regarding DVT risk factors.

In Paragraph 5.6 repetitions regarding DVT risk factors were corrected.

Paragraph 5.7: High Altitude Cerebral Edema=HACE and NOT HACO!!!

The manuscript has been thoroughly corrected of all errors.

In general I would better separate physiological and physiopathological parts from clinical and therapeutical advices, also in this case using tables.

We have re-arranged the text in such a way that physiological and pathophysiological parts were divided from clinical topics in each paragraph and clinical recommendations have been also placed in the suitable table.

Reviewer 2 Report

Review report for «The impact of temporary stay at high altitude on the circulatory system»

Summary

Mikolajczak et al. provide a comprehensive, narrative review about the impact of acute hypobaric hypoxia in patients with cardiovascular diseases travelling to high altitude. They cover several diseases such as arterial hypertension, pulmonary hypertension, heart failure, coronary artery disease, arrythmias,periperhal circulatory disorders and ischemic stroke, furthermore, they discuss possible preventive measures to prevent acute mountain sickness and disease-specific worsening of symptoms during a stay at high altitude.

I read the review with interest and I appreciate the hard work behind this review. Unfortunately, I feel that the burden of covering all cardiovascular diseases and potential treatments were large and due to this burden, highly relevant details were missed or not discussed enough. Therefore, there are several major concerns with should be addressed before publication.

Major comments

  • There is no method section. Therefore, it is completely unclear how the review was conducted, type of review, registration of review, what search terms were used, what studies were included ect. I do not request to stick to the PRISMA guidelines, since it seems that this is a narrative review (not a systematic review), however, it is mandatory, that a method section is provided with the relevant information for the readers.
  • The references are not covering the up-to-date research and I have made many comments on them. It would be important to review the references and cover them appropriately. I do understand that across all these topics, it might be challenging, however, the authors might consider to delete one or two sections to improve others. Also from my point of view, I am not an expert in all topics, therefore, I might have missed some.
  • The review article includes many small and inconclusive studies in various fields, therefore, I would recommend that some conclusions drawn by the authors should be based on the evidence level and availability of randomized clinical trials (and there are not many!). Otherwise, please let the readers know that the evidence is poor and that more research is necessary to make any recommendations.
  • The Introduction and Section 2 and 3 could be merged. Then the review would give the background in healthy subjects adapting to high altitude. Since the authors cover only a few adaptations in healthy, they should refer to reviews or books (High Altitude Medicine and Biology) which provide the full picture in healthy going to high altitude. They could end the introduction that such reviews are not available for patients with cardiovascular diseases, why the aim of this article was to fill our gap of knowledge.
  • Generally, you discuss preventive measures, however, you do not go into detail about the underlying evidence levels. There are certainly only a few randomized clinical trials available in patients with pre-existing diseases going to altitude. Therefore, you should soften your recommendations and conclusions based on the available evidence level.

Minor comments

Introduction

  • Line 26: to 27. This sentence is confusing. I understand that 4.543 billion passengers chose the plane to go to high altitude. Please rephrase this sentence.
  • Line 28: “The planes fly at an altitude of over 10000m”, but the cabin pressure is lower. Please give the reader the information that the cabin-pressure is pressurized to XY and XY meters and add a reference.

Differences in low and high-altitude environmental conditions

  • Line 47 to 48. “….and oxygen partial pressure 21%. At high altitude, these parameters reach different values”. This statement is wrong. The fraction of oxygen in the air is 21% and remains constant, however, the inspiratory partial pressure of oxygen (reported in mmHg or kPa) decreases with the reduction in barometric pressure. Please rephrase this sentence.
  • Line 49: “The barometric pressure drops more and more rapidly with increasing altitude”. This statement is wrong, the barometric pressure does not drop “more and more” with altitude. It is more a hyperbolic decrease.  

Lungs at altitude – altitude sickness

  • It is not clear to me why ventilatory adaptation to high altitude and AMS/HACE/HAPE are combined in one section. You could consider to dedicate a section to altitude-related illnesses
  • Line 114: please use the term “acute mountain sickness” instead of “altitude sickness”. “Altitude sickness” covers many different diseases, such as chronic altitude sickness, high altitude pulmonary hypertension ect.
  • “It applies to both sick and those conserved healthy”. Please add references.
  • Reference 23 is outdate, please replace it with either Luks AM, Auerbach PS, Freer L, et al. Wilderness Medical Society Practice Guidelines for the Prevention and Treatment of Acute Altitude Illness: 2019 Update. Wilderness Environ Med. 2019. OR Meier D, Collet TH, Locatelli I, et al. Does this patient have acute mountain sickness? The rational clinical examination systematic review. JAMA. 2017;318(18):1810-1819
  • I was not able to open Ref 24. Please update the reference…If possible an original article in English.
  • Update reference 25 and 26 and for the effects of sleep, cognition and sleep-disordered breathing add the reference Bloch KE, Buenzli JC, Latshang TD, Ulrich S. Sleep at high altitude: guesses and facts. J Appl Physiol. 2015;119(12):1466-1480.
  • Line 121 to 123, references for HAPE HACE and AMS: please reference Bärtsch P, Swenson ER. Acute high-altitude illnesses. New England Journal of Medicine. 2013;368(24):2294-2302.
  • Line 122: “The cause is capillary leakage in the brain (AMS and HACE) or lungs (HAPE)”. That is certainly not the only pathophysiological cause of the diseases. Actually, the pathophysiology is not well understood and there are several reviews covering each topic. Please search the literature and add the relevant references. Pathophysiology of altitude-related illnesses are out of scope of this review.
  • Line 124, replace “mountain sickness” with “acute mountain sickness”.
  • Line 123 to 127. When discussing Prevention and Treatment of altitude-related illnesses, please refer to Luks et al 2019 Wilderness Guidelines. Furthermore, adjust your recommendations based on the available evidence-level provided by Luks et al.

Arterial hypertension

  • First sentence, add a reference for the 30% of the adult population
  • Please replace reference 11 with the original article.
  • Line 150. Please add and consider the findings of this study: Bilo G, Caldara G, Styczkiewicz K, et al. Effects of selective and nonselective beta-blockade on 24-h ambulatory blood pressure under hypobaric hypoxia at altitude. J Hypertens. 2011;29(2):380-387.

Pulmonary hypertension

  • Introduce the abbreviation NYHA (maybe I have missed the introduction).
  • Line 167 and discussion about cabin pressure, please consider this reference: Ahmedzai S, Balfour-Lynn IM, Bewick T, et al. Managing passengers with stable respiratory disease planning air travel: British Thoracic Society recommendations. Thorax. 2011;66(9):831-833.
  • The rationale why PH patients are at increased risk during hypobaric hypoxia is at the end of this section. I would rearrange the section so it is clear what the risk in PH patients going to altitude is

Heart failure

  • To stay consistent, replace “height” with “altitude” throughout the article.
  • Line 181 to 184: Please be cautious about the exercise decline with altitude in healthy. See Fulco CS, Rock PB, Cymerman A. Maximal and submaximal exercise performance at altitude. Aviat Space Environ Med. 1998;69(8):793-801., they propose another decline....beginning at 700m, a decline of 8% for every additional 1000m of altitude.
  • End of sentence at ling 186. Add the reference from Furian M et al Furian M, Hartmann SE, Latshang TD, et al. Exercise performance of lowlanders with COPD at 2,590 m: data from a randomized trial. Respiration. 2018;95(6):422-432.
  • Line 196. Also add and discuss this position paper: Ahmedzai S, Balfour-Lynn IM, Bewick T, et al. Managing passengers with stable respiratory disease planning air travel: British Thoracic Society recommendations. Thorax. 2011;66(9):831-833.
  • Line 206. Please consider and discuss the findings of this paper: Bilo G, Caldara G, Styczkiewicz K, et al. Effects of selective and nonselective beta-blockade on 24-h ambulatory blood pressure under hypobaric hypoxia at altitude. J Hypertens. 2011;29(2):380-387.

Arrythmias

  • There are also data on QT prolongation at high altitude. However, the clinical significance of QT prolongation of a few ms is unclear. Further reference: Carta AF, Bitos K, Furian M, Mademilov M, Sheraliev U, Marazhapov NH, Lichtblau M, Schneider SR, Sooronbaev T, Bloch KE, Ulrich S. ECG changes at rest and during exercise in lowlanders with COPD travelling to 3100 m. Int J Cardiol. 2021 Feb 1;324:173-179. doi: 10.1016/j.ijcard.2020.09.055. Epub 2020 Sep 25. PMID: 32987054.

Peripheral circulatory disorders

  • Ref 52 is in Polish, is there an English version available?
  • Line 312 is it truly 0.28%?
  • Line 314 a “.” Is missing.
  • Line 322: “30,48 m. Is this a mistake?

Ischemic stroke

  • Replace HACO with HACE. You have already introduced this term.

Conclusion

  • Line 249: replace mountain sickness with “acute mountain sickness”
  • In these patients not only AMS is of clinical relevance…Disease-specific worsening are highly important and should be highlighted. We lack today knowledge whether low altitude treatments are effective in preventing altitude-related worsening of disease-specific symptoms. I think this should be mentioned in the conclusions.

Author Response

Answering reviewers and their precious and constructive comments we have attempted to re-arrange body of the text taking all points into account. The manuscript has been thoroughly corrected of all errors and new suggested parts have been included, also we have attached the suitable references. Also, some unclear sentences were erased or replaced by the new ones addressing the problem properly. Major changes are listed below:

There is no method section. Therefore, it is completely unclear how the review was conducted, type of review, registration of review, what search terms were used, what studies were included ect. I do not request to stick to the PRISMA guidelines, since it seems that this is a narrative review (not a systematic review), however, it is mandatory, that a method section is provided with the relevant information for the readers.

We have added a method section in the proposed manner.

The references are not covering the up-to-date research and I have made many comments on them. It would be important to review the references and cover them appropriately. I do understand that across all these topics, it might be challenging, however, the authors might consider to delete one or two sections to improve others. Also from my point of view, I am not an expert in all topics, therefore, I might have missed some. The review article includes many small and inconclusive studies in various fields, therefore, I would recommend that some conclusions drawn by the authors should be based on the evidence level and availability of randomized clinical trials (and there are not many!). Otherwise, please let the readers know that the evidence is poor and that more research is necessary to make any recommendations.

References were checked and corrected. We merged some sections and significantly re-arranged the other ones attempting to make the text more clear and understandable. As suggested by the Reviewer some studies cited were inconclusive and there is still a lack of randomized trials in this area – that is why at the ending we introduced a small paragraph on the limitations explaining the strength of some recommendations.

The Introduction and Section 2 and 3 could be merged. Then the review would give the background in healthy subjects adapting to high altitude. Since the authors cover only a few adaptations in healthy, they should refer to reviews or books (High Altitude Medicine and Biology) which provide the full picture in healthy going to high altitude. They could end the introduction that such reviews are not available for patients with cardiovascular diseases, why the aim of this article was to fill our gap of knowledge.

We have added sentences underlining that such reviews are not available for patients with cardiovascular diseases, and that the aim of this article was to fill the gap of knowledge.

Generally, you discuss preventive measures, however, you do not go into detail about the underlying evidence levels. There are certainly only a few randomized clinical trials available in patients with pre-existing diseases going to altitude. Therefore, you should soften your recommendations and conclusions based on the available evidence level.

We have re-arranged the text in such a way that physiological and pathophysiological parts were divided from clinical topics in each paragraph and clinical recommendations have been also placed in the suitable table.

Minor comments

Numerous suggested minor changes has been done including rephrasing of some sentences and attaching new references.

Round 2

Reviewer 1 Report

I appreciate the authors' effort in radically revising the text by correcting the suggested errors. In particular, the addiction in each paragraph of a specific section dedicated to recommendations better directs the work to a larger number of doctors who want to advise their patients. I found the revised version more balanced between physiological and clinical parts. Three more minor corrections in the text: 1) at page 1 line 38 altitude and not attitude; 2) in table 2 right side penultimate point the end of the sentence is not clear "during flights predispose to....?"; 3) page 9 line 420 pregnancy and puerperium are repeated as in the previous line.

Author Response

Dear Reviewer,

Thank you for careful and thorough reading of this manuscript and for the thoughtful comments. We are incredibly grateful for the review.

Comment of Reviewer: At page 1 line 38 altitude and not attitude.

Changes carried out in the paper: The term "attitude" has been replaced by the term "altitude".

Comment of Reviewer: In table 2 right side penultimate point the end of the sentence is not clear "during flights predispose to....?"

Changes carried out in the paper: The imprecise, mistakenly introduced wording "during flights predispose to…" has been deleted.

Comment of Reviewer: page 9 line 420 pregnancy and puerperium are repeated as in the previous line.

Changes carried out in the paper: The repetition of "pregnancy, puerperium" has been deleted.

Best regards,

Authors

Reviewer 2 Report

R2 : Review report for «The impact of temporary stay at high altitude on the circulatory system»

I thank the authors for the revised version of the manuscript. It has improved in structure, references and conten, but has still some major flaws in the structure and quality of the manuscript.

The manuscript has several writing errors and I strongly recommend the authors to check the spelling and grammar from an English Native.

Additionally, I request that the authors check the manuscript for redundancy. As examples:

Line 120 to 128 covers the fact that at altitude the barometric pressure and PaO2 decreases. This information is also described between lines 57 to 67.

The section between line 138 to 147 describes the purpose of this article, however, the physiology in healthy was described before this section, which is confusing. Please rearrange the manuscript:

  1. Introduction (general introduction) + ending the section with the purpose of the article.
  2. Method (please add this heading to the article)

Headings, please begin here with

  1. Physiology and Pathophysiology in healthy at high altitude. Subheadings should be normal physiological adaptation (please describe the adatpations in the order of their appearance); AMS.
  2. Then the sections describing each patient population

Line 27: “Moreover, every year lots of….” This sentence is strange and makes no sense, please rephrase this sentence. In an airplane you are not at very high altitude, since the cabin pressure is pressurized to 1800 – 2438 meters.

Line 38: altitude

Line 57 to 58. “….and oxygen partial pressure 21%.” Please change to “….and the fraction of oxygen in the air is 0.21.

Next sentence: Change to “With high altitude, the barometric pressure decreases and results in lower levels of oxygen partial pressure”.     

…..”

Table 2: Please rearrange the physiological changes in the order of their appearance.

Line 95: Change from “This is due to their narrowing under the influence of hypoxia” to “This is due to hypoxic pulmonary vasoconstriction”.

Line 120, delete the first sentence due to redundancy. Furthermore, this section describes hyperventilation in response to hypoxemia. This response is one of the first physiological adaptations and this section should be earlier in the  manuscript. Generally, please rearrange the sections covering the physiological responses to the order their appearance when exposed to high altitude..

Line 129 to 137: Here is assume that you added a section about methods, however, no heading is provided… Are these the methods or not? If yes, please add a proper heading.

Section line 138 to 147: This section should be at the end of the introduction and before the Methods. Please rearrange.

Section AMS, line 153 to 156: Dyspnea and other described symptoms here are not symptoms of AMS. Since this section covers AMS, please stick to AMS and do not dilute the section. Additionally, you describe HACE and HAPE… Either you change the heading to acute high altitude illnesses. Or you rearrange the section. Please stay within the topic of the heading.

Additionally, the definition of AMS has changed (see Roach Lake Louise Consensus 2018), therefore, sleep problems are not part of AMS anymore. You have to carefully check this section, since there are several mistakes misleading readers not experienced with AMS. 

Line 283: This publication was not done by Falco et al. Change to Furian et al. I have found this mistake just by chance, pelase check whether there are other similar mistakes in the manuscript.

Line 320 to 322: I cannot believe this statement. The supply of the heart muscle is maintained by expanding the coronary arteries?! From where is this information? The oxygen supply is maintained due to hyperventilation, increased cardiac output by an increase in heart rate and erythropoiesis…if there is an increase in diameter of the arteries, then this is at very high altitude; a marginal effect not really relevant for the oxygen supply of the heart.

Section at Line 378: Carta et al. do not show a shortening of QTc with altitude. This is wrong. Please check again and change the sentence.

Line 382: “Both studies were conducted in patients with moderate to severe COPD”. Your statement that the studies were conducted on “a group” of COPD patients let the reader think that the two studies were done in the same patients. This is wrong, the studies were conducted at different places with different COPD patients.

Line 389: Change the word complexed, this word is strange.

Table 3: I do not know which treatments have been studied in RCTs, which ones in case-control studies, case series ect… As a reader, I do not know which recommendations are evidence-based and which one not. Please help the reader to properly use the table.

Line 295: Change HTC to HCT